# Always Online? Internet Addiction and Social Impairment in Psoriasis across Germany

**DOI:** 10.3390/jcm9061818

**Published:** 2020-06-11

**Authors:** Maximilian Christian Schielein, Linda Tizek, Barbara Schuster, Stefanie Ziehfreund, Claudia Liebram, Kilian Eyerich, Alexander Zink

**Affiliations:** 1Department of Dermatology and Allergy, School of Medicine, Technical University of Munich, 80802 Munich, Germany; linda.tizek@tum.de (L.T.); barbara.schuster@tum.de (B.S.); stefanie.ziehfreund@tum.de (S.Z.); kilian.eyerich@ki.se (K.E.); 2Biometry and Epidemiology (IBE), Department of Medical Informatics, Ludwig-Maximilians-University, 81377 Munich, Germany; 3Pettenkofer School of Public Health, Ludwig-Maximilians-University, 81377 Munich, Germany; 4Psoriasis-Netz e.V., 13437 Berlin, Germany; redaktion@psoriasis-netz.de; 5Karolinska Institutet, Department of Medicine, Unit of Dermatology and Venerology, Karolinska University Hospital, 171 77 Stockholm, Sweden

**Keywords:** psoriasis, Internet addiction, people-centered care, social impairment, stigmatization

## Abstract

With the World Health Organization (WHO) demanding further investigation of the social impairment and psychosocial burden of psoriasis, a first study identified a high prevalence of Internet addiction. The aim of this study was to assess social impairment and estimate the occurrence of Internet addiction along with depression, cigarette smoking, and alcohol dependency in individuals with psoriasis recruited online in a people-centered care approach. A cross-sectional online survey was carried out across Germany between March 2019 and June 2019. The questionnaire contained information on social impairment, smoking habits, as well as validated questionnaires on Internet addiction, depression, and alcohol dependency. Overall, 460 individuals (62.4% female; mean age: 45.9 ± 13.7 years) with psoriasis were included. Of those, 406 (88.3%) stated to be at least rarely socially impaired. The positive screening rate for Internet addiction was 8.5%. Furthermore, 40.0% had positive screenings for depression, 17.1% for alcohol dependency, and 32.6% for daily smoking. Positive screenings for Internet addiction and alcohol dependency were substantially more frequent in individuals with psoriasis than in the German general population. In order to meet the demands of the WHO, Internet addiction could be considered as a potential comorbidity in psoriasis and a focus on people-centered care is advisable for further research.

## 1. Introduction

Internet addiction is a phenomenon first appearing at the turn of the millennium that has since then begun to rise immensely in importance [1]. It was classified as the most potent problem within the revised Diagnostic and Statistical Manual of Mental Disorders (DSM) in 2013 [2]. In Germany, prevalence estimations range from 1.0% in the general population up to 3.2% in the subgroup of adolescents [3,4]. With regard to skin diseases, a recent study found that pathological Internet use and Internet addiction were substantially more frequent among a sample of 502 patients with psoriasis (3.8%) than in the general population (1.0%) [3,5].

Psoriasis is a chronic inflammatory skin disease affecting 1.2–3% of individuals in Germany [6,7,8]. Patients often have an impaired quality of life and reduced happiness [8,9]. Individuals with psoriasis tend to avoid or reduce physical activities and often withdraw themselves from social activities [10,11] and intimate contact [12,13]. These social impairments are broadly individual for each patient and associated with various comorbidities [14]. Patients often suffer from a psychosocial burden due to stigmatization [15,16] as well as from comorbidities such as depression and addictions [17,18]. Subsequently, the World Health Organization (WHO) emphasized the importance of recognizing the stigmatization in psoriasis and its potential consequences for burden of disease and mental comorbidities [19]. While reviews indicate more frequent alcohol dependency and cigarette smoking in psoriasis patients [20,21], evidence on compulsive Internet use and Internet addiction remains limited to one study [5].

If treated appropriately, patients with psoriasis can benefit from highly effective therapies. Individuals treated effectively tend to have not only less severe skin lesions but also reduced depressive symptoms and social impairment [22,23]. The reduction of comorbidities and the promotion of mental health of individuals with psoriasis are essential according to the WHO [19]. However, not all patients receive therapies as recommended by guidelines [24,25], and, since not all affected individuals seek medical care, many affected individuals are not considered as psoriasis patients [26]. Hence, the WHO demands to focus on people-centered instead of patient-centered health care [27,28]. Despite this, most research still focuses on registries including mainly, moderately, and severely affected individuals and typical patient settings such as dermatological practices and clinics. To reach individuals outside of conventional settings, online approaches can be beneficial [13,29] as many individuals, regardless of contact to a physician, search for health-related information online [30,31,32,33].

The aim of this study was therefore to assess social impairment and to estimate the occurrence of Internet addiction along with depression, smoking, and alcohol dependency among individuals with psoriasis using a people-centered online approach.

## 2. Experimental Section

### 2.1. Study Design and Recruitment

This cross-sectional study was carried out as an online survey among individuals with psoriasis in Germany from March to June 2019. The online questionnaire was distributed via an online self-help platform “Psoriasis-Netz”, a patient online platform “Farbenhaut” as well as a national campaign of the “Association of the German Dermatologists” (BVDD). The project was most promoted by “Psoriasis-Netz” on their website together with current information for individuals interested in psoriasis. Its monthly e-mail newsletter was sent to 2296 registered individuals across Germany. “Farbenhaut” and the BVDD shared the questionnaire on their social media channels one month before completion of recruitment for the study.

Only individuals who stated having psoriasis diagnosed by a physician were included in the analyses. Additional exclusion criteria were the presence of implausible data or more than 20% of missing values. Electronic informed consent from each participant was acquired prior to study inclusion. All study procedures were in accordance with the Declaration of Helsinki and were reviewed as well as approved by the local ethics committee of the Technical University of Munich (reference 25/19 S).

### 2.2. Questionnaire

The study questionnaire was developed in collaboration with “Psoriasis-Netz”. One dermatologist, two epidemiologists, and two members of “Psoriasis-Netz” were involved in the process. Questions were only added if they were accepted unanimously. The questionnaire was pre-tested by three researchers and three individuals affected by psoriasis and adapted according to their comments.

Participants answered questions on sociodemographic variables and their medical history, including age, gender, disease severity in general and at time of participation (self-classification as “mild”, “moderate”, or “severe”, respectively) as well as disease duration and current utilization of medical care. Due to the nature of the chosen online approach, standardized reflection of disease severity using physician-based Psoriasis Area and Severity Index (PASI) or body surface affected (BSA) was not possible. In order to keep the questionnaire concise, social impairment was assessed with a one-question item asking “Does your psoriasis prevent you from pursuing certain leisure activities?”, which could be answered on a five-point Likert scale ranging from “never” to “always”. Participants who stated that their psoriasis prevents them from taking part in certain leisure activities were asked for the main restrictions using free-text comments. After revising the questionnaire, examples for possible answers were added (“e.g., swimming, sauna, sunbathing, …”).

Internet addiction was assessed using the Compulsive Internet Using Scale (CIUS; Cronbach’s α = 0.93) [34,35]. The questionnaire comprises 14 questions, which are to be answered on a five-point Likert-scale ranging from “never = 0” to “very often = 4”. Subsequently, the score ranges from 0 to 56. A cutoff score of 21 was used to estimate the prevalence of Internet addiction [5,35]. Additionally, participants were asked to state the days per week and hours per day spent online in their leisure time.

Depressive behavior was assessed with the International Classification of Diseases (ICD)-10-based WHO-Five Well-Being Index (WHO-5, Cronbach’s α = 0.88) [36,37], a widely used, validated questionnaire comprising five questions on well-being. Answers range from “never” to “always” and are rated from zero to five, respectively. The resulting sum is multiplied by four, resulting in a score between 0 and 100. A cutoff value of ≤28 for depression showed a sensitivity of 0.94 and a specificity of 0.83 and, therefore, was used to determine depression as a study outcome [38].

To screen for alcohol use disorder, the DSM-based CAGE-questionnaire was used [39]. It comprises questions on “cutting down”, “annoyed by criticism”, “guilt about drinking”, and alcohol as an “eye-opener” in the morning. Questions are answered with “no” or “yes”. The subsequent score ranges from zero to four. The instrument showed good test-retest reliability (0.80–0.95) and the cutoff value of at least two questions answered with “yes” as a positive screening for alcohol use disorder previously showed a sensitivity of 0.71 and a specificity of 0.90 [40].

Smoking was assessed by one question: “Do you smoke cigarettes?”. Participants who stated that they “never” or “seldom” smoke were classified as non-smokers. Participants who stated that they smoke daily, regardless of the stated amount, were considered smokers.

### 2.3. Statistical Analyses

As the online method of patient recruitment of this people-centered survey was explorative, study size was determined by the number of individuals recruited during a three-month time frame. Descriptive data were computed for all participants and stratified by social impairment. Group differences were calculated using unpaired t-tests or chi-square tests. Prevalence of positive screenings for Internet addiction along with those for depression, smoking, and alcohol dependency were determined. Results were stratified by gender, age (by median split; 46 years), and social impairment. To avoid confounding, possible influencing factors were assessed using univariate and multiple regression models. All factors that showed a significant association in the univariate analysis, were entered in the multiple regression model with backward selection. Independent variables included age, gender, disease duration, utilization of medical care, disease severity at time of study participation, severity in general, and social impairment. Odds ratios (OR) and respective 95% confidence intervals (95% CIs) were calculated. To analyze activities avoided due to psoriasis, free-text answers were categorized using an inductive analyzing procedure. Categories were descriptively quantified. Additionally, quantities of the 50 most commonly used words were visualized while excluding nonspecific words such as “I”, “with”, or “do”. IBM SPSS Statistics (Version 25, IBM Corporation, Armonk, NY, USA) was used for all analyses and alpha was set at 0.05.

## 3. Results

A total of 466 individuals with psoriasis participated in this study. Of these, six were excluded due to implausible data, resulting in a total of 460 participants being analyzed. The mean age was 45.9 ± 13.7 years and 62.4% (*n* = 287) of the participants were female. The mean duration of disease was 21.0 ± 14.7 years and 22.8% (*n* = 105) of the participants were currently not in medical care. About half of the participants stated to have a moderate disease severity both at the time of study participation (55.0%; *n* = 253) and in general (56.3%; *n* = 259). When comparing general and current disease severity, 32.8% (*n* = 151) of the participants stated that their psoriasis was better at the time of study participation, while 13.0% (*n* = 60) indicated a worse disease severity (Table 1).

### 3.1. Social Impairment

Overall, 330 (71.7%) stated that their psoriasis at least sometimes prevents them from certain leisure activities. Of these, 124 (27.0%) individuals answered this question with “frequently” and 92 (20.0%) with “always”. Participants indicating that their psoriasis prevents them at least sometimes from certain leisure activities reported a higher rate of severe disease characteristics at the time of study participation (29.1% vs. 6.9%; *p* < 0.001) and in general (42.1% vs. 19.2%; *p* < 0.001) than participants who indicated no or rare impairment (Table 1).

Furthermore, 406 (88.3%) participants stated that their psoriasis at least rarely prevents them from any leisure activity. When asked which leisure activities were impaired with an open question, 394 participants (97.0%) provided 552 answers. After qualitatively categorizing all given answers, “swimming” (*n* = 273; 67.2%; e.g., “Swimming in public pools. The chlorine burns the skin.”), “sport” (*n* = 93; 22.9%; e.g., “I can’t go jogging anymore as my knees were destroyed by psoriatic arthritis.”), and “stigmatization and appearance” (*n* = 49; 12.1%; e.g., “Whenever I can’t put on anything long enough to hide my disease in public”) were the three most mentioned categories (Table 2). Furthermore, many answers indicated a reduction in social contacts. For example, people used expressions such as “meeting new people” and “any activity among people is unpleasant because the strong itching leads to scratching and leaving dandruff everywhere”. Many individuals also mentioned abstaining from activities that could exacerbate their symptoms, such as “drinking/eating/partying—all because of the fear of worsening condition following the consumption of unhealthy food/alcohol”. Reasons given widely differ within the individuals and the 50 most commonly used words show a multifaceted sense of loss and preoccupation with daily life for participants due to psoriasis (Figure 1).

### 3.2. Internet Addiction

Participants reported spending 21.6 ± 12.5 h per week online excluding time at work. About four out of five participants stated being online for private reasons every day (*n* = 381; 82.8%). Overall, 8.5% (*n* = 39) of all participants were screened positive for Internet addiction, with no significant difference in gender (female: 8.5% vs. male: 8.7%, *p* = 0.935), age (<46 years: 9.1 vs. ≥46 years: 8.0%, *p* = 0.660), and social impairment due to psoriasis (“Never or rarely”: 8.5% vs. “Sometimes, frequently, or always”: 8.5%, *p* = 0.997; Figure 2, Table A1).

### 3.3. Depression, Smoking, and Drinking

Depressive tendencies were found in 40.0% (*n* = 180) of the participants (Figure 1). Social impairment due to psoriasis (at least “sometimes”) was associated with a higher proportion of positive screening results for depression (45.8% vs. 25.2%; *p* < 0.001). Furthermore, 32.6% (*n* = 150) of all individuals stated to smoke cigarettes daily, and 17.1% (*n* = 77) were screened positive for alcohol addiction. More women reported a daily smoking habit (38.3% vs. 23.1%; *p* = 0.001; Figure 1), while more men were screened positive for alcohol addiction (25.9% vs. 11.8%; *p* < 0.001). Additionally, younger participants more frequently reported smoking cigarettes daily (41.4% vs. 24.4%; *p* < 0.001). These differences remained significant when controlled in a multiple regression model, resulting in ORs of 2.13 (95% CI: 1.36; 3.34) for women and 0.96 (95% CI: 0.95; 0.98) for age (Table A2).

## 4. Discussion

This study aimed to characterize social impairment and estimate the occurrence of Internet addiction along with depression, smoking, and alcohol dependency in individuals with psoriasis recruited via an online, people-centered care approach. Many participants indicated an impairment due to their psoriasis, with swimming and sports being the most commonly mentioned fields of daily life being avoided. Given reasons often focused on stigmatization and pain. Furthermore, a high positive screening rate for Internet addiction and alcohol dependency was found.

### 4.1. Social Impairment

Overall, 88.3% of participants indicated that their psoriasis prevents them from leisure activities and meeting other people. Thereby, individuals with more social impairment reported higher self-perceived disease severity. This finding is in line with a previous study, which found that individuals with moderate or severe disease severity engaged approximately 30% less in leisure activities than healthy controls did; no difference was observed for participants with little or no disease activity [10]. In our study, we not only found that people avoided specific leisure activities such as swimming and sports but that they also felt stigmatized. This confirms previous findings that stigmatization of skin lesions was associated with social impairment [16]. Most of the participants mentioned avoiding swimming. However, it should be noted that examples for leisure activities such as “Swimming, sauna, sunbathing, …” were provided as suggestions to give participants ideas for possible answers, potentially resulting in biased free-text answers. The high number of mentions for swimming, however, reflect a problem identified decades ago [41]. In 1989, a study on 104 psoriasis patients found that 72% of patients avoided swimming, 60% avoided sunbathing, and 40% avoided sports. Although sunbathing was also mentioned as an example in this study, it was mentioned less frequently in this sample, while swimming remains an often avoided activity for affected individuals [41]. The stated reasons of shame, stigmatization, and physical sensations such as burning and itching are in line with previous literature [10,41]. The fact that patients still abstain from activities such as swimming because of possible stigmatization emphasizes the importance of current efforts to reduce stigmatization in psoriasis [15,16], which follows the call for action outlined by the WHO [19]. Another point that might be addressed in future research is a possible connection between skin and joint pain as well as itch, and social impairment. Skin pain is an often prevalent symptom [42] and was also frequently reported in the free-text answers.

### 4.2. Internet Addiction

Positive screening rates for Internet addiction exceeded those reported in the literature [3,4,5,43]. In comparison to a German representative study among adolescents, this study’s results were considerably higher (8.5% vs. 3.2%) [4]. This is surprising, considering the difference in mean age (45.9 years vs. 14.9 years) and tendency for Internet addiction to occur in younger individuals and to decrease with age [3]. Even if we were to consider a higher cutoff value of 28, our detected prevalence of Internet addiction was higher (3.2%) than the prevalence of 1.0% in another German study among 8132 adolescents and adults (mean age: 39.9 years) [3]. Although a third study also used online recruitment via Facebook groups, they only detected a prevalence of 1.2% of Internet addiction in 245 regular Internet users [43]. Lastly, the prevalence of positive screening results also outnumbered the prevalence among 502 psoriasis patients recruited from various dermatological practices and clinics throughout southern Germany (8.5% vs. 3.6%) [5]. Since our results on Internet addiction exceed all previous reported numbers, this could imply that by recruiting participants online and via patient platforms as well as a nationwide physician-lead campaign, we were able to reach a highly vulnerable subgroup of individuals with psoriasis. As another study by Megna et al. found higher signs of inflammation in patients with psoriatic arthritis who practice smartphone overuse [44], future studies should include patient stratification by presence of psoriatic arthritis and differentiation between online and smart phone addiction. This is, however, one of the first studies to investigate Internet addiction in psoriasis and further investigation should follow.

### 4.3. Depression, Smoking, and Alcohol Dependency

Our detected value for positive screenings for depression is similar to one of the highest prevalence rates reported by a systematic review on depression in patients with psoriasis (questionnaire-based prevalence: 13.8–39.2%) [17]. The high rate of depression in our study might be explained by the fact that a people-centered care approach was used instead of a patient-centered approach. This could be beneficial in reaching especially vulnerable subgroups of affected individuals. In accordance with preliminary studies in patients with psoriasis [5,18,20,21] and compared to a representative German sample, individuals with psoriasis reported a higher prevalence of daily smoking (32.6% vs. 15.1%) [45] and alcohol dependency (17.1% vs. 3.1%) [45]. The gender distribution for smokers was contrary to that of the general population in which men are more likely to smoke daily than women [45]. Positive screenings for alcohol dependency (17.1%) also exceeded values reported in two recent German studies in patients with psoriasis (8.6–13.5%) [5,18]. Possible explanations might include the anonymous environment of an online survey, a more vulnerable sample in this study, or both.

### 4.4. Limitations and Strengths

There are some study limitations. As this was an online survey, the truthfulness of participant answers cannot be verified and prevalences were estimated using screening tools, not diagnoses. Additionally, due to the anonymity provided by the online design of this study, social desirability bias could have been reduced. While this is desirable, it makes comparison with previous studies in medical settings more difficult. Selection bias must be taken into account when considering the generalizability of these findings. Mainly individuals receiving information, newsletters, or social media updates from the multiplier institutions were reached. Individuals who are not engaging with online content related to psoriasis, who are participating in other organizations, or who do not have an Internet connection were highly unlikely to participate in our study. However, this might have also led to an especially vulnerable subgroup of affected individuals, which can be desirable when evaluating mental and social impairment. Those with a high disease burden might be more likely to search for further information online and therefore have may have a higher chance of receiving a study invitation through a multiplier organization. Since a high proportion of participants in this study were not currently in medical care, this online-based recruitment strategy allowed us to reach a unique group of individuals, who may not have been considered in traditional clinical trials and registries, further showing the strengths of online outreach. This can broaden the horizon of dermatological research and strengthen people-centered care [29].

## 5. Conclusions

The study implicates that social impairment and Internet addiction are high among individuals with psoriasis recruited via patient networks in a people-centered care and online approach. Positive screening results for Internet addiction and the other mental health variables exceeded values found for the general population. The findings on social impairment and addictions emphasize the importance of mental burden in psoriasis [19] and, therefore, strengthen evidence on the need for programs to reduce stigmatization [15,16]. Internet addiction was confirmed as an aspect of mental health that should be considered in further research on individuals with psoriasis. Ultimately, the results indicate that inclusion of online self-help platforms and their users in health care research could be a key element in promoting people-centered and not only patient-centered care.

## Figures and Tables

**Figure 1 jcm-09-01818-f001:**
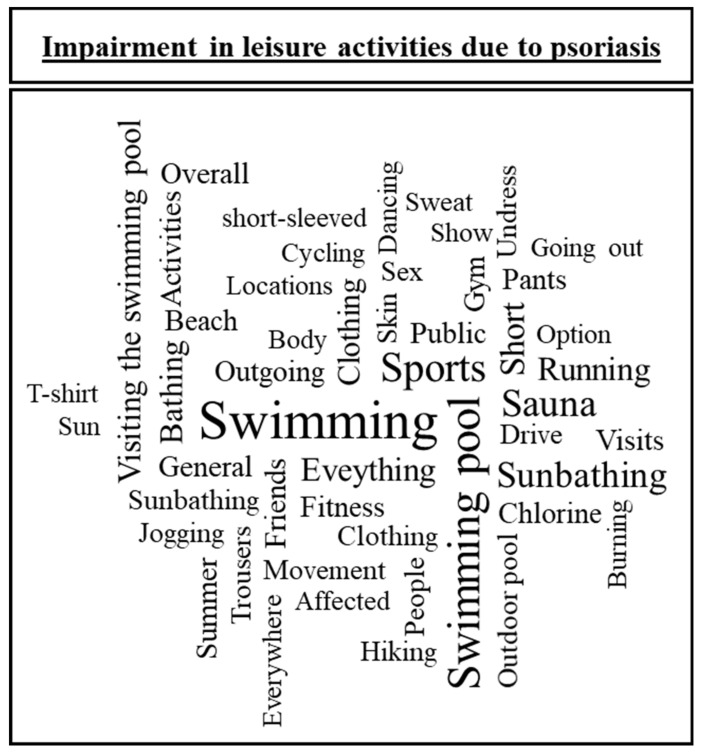
The 50 most common words in free-text answers on the question, which leisure activities were prevented by psoriasis. Words were ranked by frequency. Font size (fs) equals the sweeping break of the third root of word’s rank (rx) times maximal font size (fs_max_) [fs = fs_max_*rx^−1/3^]. As some words cannot be translated verbatim and free-text answers were given in German, some words are separated in two or more or appear as duplicates.

**Figure 2 jcm-09-01818-f002:**
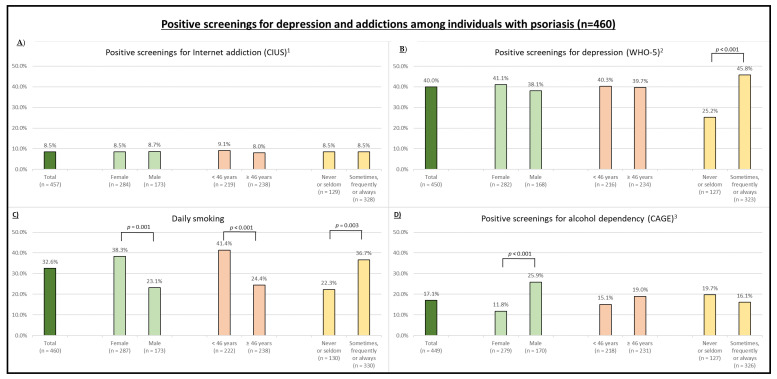
Positive screenings for (**A**) Internet addiction, (**B**) depression, (**C**) cigarette smoking, and (**D**) alcohol dependency. Positive screenings are stratified by gender, age (median split), and social impairment. ^1^ Measured using the Compulsive Internet Using Scale (CIUS; cutoff: ≥21; range: 0–56). ^2^ Measured using the World Health Organization (WHO)-Five Well-Being Index (WHO-5) questionnaire (cutoff: ≤29; range: 0–100). ^3^ Measured using the CAGE questionnaire (cutoff: ≥2; range: 0–4).

**Table 1 jcm-09-01818-t001:** Characteristics of study participants in total and stratified by influence of psoriasis on avoiding free-time activities.

	Total (*n* = 460)	Psoriasis is Preventing Leisure Activities		*p*-Value
		Never or Rarely (*n* = 130)	Sometimes, Frequently, or Always (*n* = 330)	
Age (years)				
(Mean, SD)	45.9 ± 13.7	46.7 ± 14.1	45.6 ± 13.6	0.426
Age group <46	222 (48.3%)	62 (47.7%)	160 (48.5%)	0.878
Age group ≥46	238 (51.7%)	68 (52.3%)	170 (51.5%)	
Gender; *n* (%)				
Female	287 (62.4%)	86 (66.2%)	201 (60.9%)	0.296
Male	173 (37.6%)	44 (33.8%)	129 (39.1%)	
Duration of psoriasis (years)				
(Mean, SD)	21.0 ± 14.7	21.9 ± 15.1	20.7 ± 14.6	0.416
Currently in medical care; *n* (%)				
Yes	355 (77.2%)	95 (73.1%)	260 (78.8%)	0.189
No	105 (22.8%)	35 (26.9%)	70 (21.2%)	
Severity at the time of study participation; *n* (%)				
Mild	102 (22.2%)	47 (36.2%)	55 (16.7%)	<0.001
Moderate	253 (55.0%)	74 (56.9%)	179 (54.2%)	
Severe	105 (22.8%)	9 (6.9%)	96 (29.1%)	
Severity in general; *n* (%)				
Mild	37 (8.0%)	22 (16.9%)	15 (4.5%)	<0.001
Moderate	259 (56.3%)	83 (63.8%)	176 (53.3%)	
Severe	164 (35.7%)	25 (19.2%)	139 (42.1%)	
Severity at study participation compared to severity in general; *n* (%)				
Worse	60 (13.0%)	12 (9.2%)	48 (14.5%)	0.225
Equal	249 (54.1%)	77 (59.2%)	172 (52.1%)	
Better	151 (32.8%)	41 (31.5%)	110 (33.3%)	

SD = standard deviation.

**Table 2 jcm-09-01818-t002:** Inductive categories of free-text answers on what leisure activities were avoided due to psoriasis. Quantity, two examples, and respective participant characteristics per category.

Category	Count *n* (%)	Example	Participant (Gender, Age)
Swimming	273 (67.2%)	“Sauna and swimming pool, but only because of the expected looks on the affected areas”	Woman, 32 years
		“Swimming in public pools. The chlorine burns the skin”.	Man, 39 years
Sport	93 (22.9%)	“I can’t go jogging anymore as my knees were destroyed by psoriatic arthritis”.	Man, 32 years
		“Running”	Woman, 51 years
Stigmatization and appearance	49 (12.1%)	“Whenever I can’t put on anything long enough to hide my disease in public”	Woman, 50 years
		“Any activity requiring short clothing”	Woman, 21 years
Sauna	32 (7.9%)	“Sauna, bathing”	Man, 73 years
		“Sauna, swimming, nudism”	Woman, 57 years
Sunbathing	30 (7.4%)	“Sunbathing at the lake”	Man, 57 years
		“Sunbathing at the beach”	Man, 30 years
Movements and walking	28 (6.9%)	“Going for a walk”	Woman, 52 years
		“Roughhousing with my son”	Man, 34 years
Going out and meeting friends	26 (6.4%)	“Any activity among people is unpleasant because the strong itching leads to scratching and leaving dandruff everywhere”.	Woman, 27 years
		“Visiting restaurants with friends”	Man, 38 years
Other	21 (5.2%)	“Everything you need hands for”	Woman, 67 years
		“Living”	Man, 38 years

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
