# Peer review of "Always Online? Internet Addiction and Social Impairment in Psoriasis across Germany"

_jcm, 2020, doi:10.3390/jcm9061818_

Round 1

Reviewer 1 Report

This is an interesting paper worth publishing. The mauscript contains some new data. The methodology is well decribed, however I will need more details on the newly created questionaire the authors used. They should provide some validation results, such as Chrombach alpha and ICC- Intraclass Correlation Coeficient. I prefer to include this questionnaire as at least e-supplement.

The main question of the study is to assess the internet addiction along with depression, smoking and alcohol dependency in patients suffering from psoriasis. The design of the study is of interest. In my opinion this question is worth investigating.

I found the methodology original. The study was condiuted via the internet. I has both advantages and disadvantages, but the authors analysed the obtained results with caution. The study definitely add new data to the current knowledge.

The paper is well written, the text  is clear and easy to read

The conclusions are consistent with the evidence and arguments
presented. They address the main question posed.

Reviewer 2 Report

“ Always online? Internet addiction and social impairment in psoriasis across German” is a well written research that focuses on emergent problems among psoriatic patients.

Comments:

  • page 9, line 279. “Authors should discuss the results and how they can be interpreted in perspective of previous studies and of the working hypotheses. The findings and their implications should be discussed in the broadest context possible. Future research directions may also be highlighted.” This is from JCM guidelines and should be cancelled from the text of the manuscript.

Reviewer 3 Report

The authors performed an online based study on internet addiction and psoriasis related impairment among psoriasis patient.

The topic is isteresting

Few points need to be addressed:

-Psoriasis limits physical activity possibilities. There are also molecular and pathogenetic links between physcial activity and psoriasis.

-Internet addiction, mainly throught samrtphone overuse, may possibly be linked to increased risk to develop psoriatic arthritis.

-It would be important to have data regarding ongoing psoriasis treatment and questionnaire outcomes. Where they better in patients under biologics? These correlations should have been done also for psoriasis severity . There were no data on ongoing PASI or BSA? Disease severity is one of the main drivers of extracutaneous impact of psoriasis, so it would be important to have more data on this topic

-Has the symptoms skin pain been investiogated? Indeed, it was previously reported that skin pain is common among psoriasis patients (Patruno C, Napolitano M, Balato N, et al. Psoriasis and skin pain: instrumental and biological evaluations. Acta Derm Venereol. 2015;95(4):432‐438), possibly driving psoriasis restriction in social, recreational and sexual life as well as in physical activity
